# Macrophage Phenotype Induced by Circulating Small Extracellular Vesicles from Women with Endometriosis

**DOI:** 10.3390/biom14070737

**Published:** 2024-06-21

**Authors:** María Angeles Martínez-Zamora, Olga Armengol-Badia, Lara Quintas-Marquès, Francisco Carmona, Daniel Closa

**Affiliations:** 1Department of Gynecology, Institut Clínic of Gynecology, Obstetrics and Neonatology, Hospital Clínic of Barcelona, Institut d’Investigacions Biomèdiques August Pi i Sunyer (IDIBAPS), 08036 Barcelona, Spain; lquintas@clinic.cat (L.Q.-M.); fcarmona@clinic.cat (F.C.); 2Department of Experimental Pathology, Institut d’Investigacions Biomèdiques de Barcelona, Consejo Superior de Investigaciones Científicas (IIBB-CSIC), Institut d’Investigacions Biomèdiques August Pi i Sunyer (IDIBAPS), 08036 Barcelona, Spain; olga.armengol@iibb.csic.es (O.A.-B.); daniel.closa@iibb.csic.es (D.C.)

**Keywords:** endometriosis, small extracellular vesicles, macrophages

## Abstract

Evidence suggests that immune system dysfunction and macrophages are involved in the disease establishment and progression of endometriosis. Among the factors involved in this alteration in macrophage activity, Small Extracellular Vesicles (sEVs) have been described to play a role favoring the switch to a specific phenotype with controversial results. This study aims to investigate the potential effect of circulating sEVs in the plasma of well-characterized patients with endometriosis on the polarization of macrophages. sEVs were isolated from the plasma of patients diagnosed with endometriosis confirmed by histopathological analysis. Two groups of patients were recruited: the endometriosis group consisted of patients diagnosed with endometriosis by imaging testing (gynecological ultrasonography and/or magnetic resonance imaging), confirmed by histopathologic study (n = 12), and the control group included patients who underwent laparoscopy for tubal sterilization without presurgical suspicion of endometriosis and without endometriosis or signs of any inflammatory pelvic condition during surgery (n = 12). Human THP1 monocytic cells were differentiated into macrophages, and the effect of sEVs on cell uptake and macrophage polarization was evaluated by fluorescent labeling and measurement of the *IL1B*, *TNF*, *ARG1*, and *MRC1* expression, respectively. Although no changes in cell uptake were detected, sEVs from endometriosis induced a polarization of macrophages toward an M2 phenotype, characterized by lower *IL1B* and *TNF* expression and a tendency to increase *MRC1* and *ARG1* levels. When macrophages were stimulated with lipopolysaccharides, less activation was also detected after treatment with endometriosis sEVs. Finally, endometriosis sEVs also induced the expression of the nuclear receptor peroxisome proliferator-activated receptor-gamma (PPARG); however, treatment with rosiglitazone, a PPARG agonist, had no effect on the change in macrophage phenotype. We conclude that circulating sEVs in women with endometriosis have a certain capacity to shift the activation state of macrophages toward an M2 phenotype, but this does not modify the uptake level or the response to PPARG ligands.

## 1. Introduction

Endometriosis is a complex, nonmalignant gynecological disorder characterized by the presence of endometrial cells outside the uterine cavity, leading to chronic pelvic pain, infertility, and a significant reduction in the quality of life for affected individuals [1]. Despite its high prevalence and impact on women’s health, the precise mechanisms underlying the pathogenesis of endometriosis remain incompletely understood [2,3,4,5,6].

Evidence suggests that immune system dysfunction is involved in the pathogenesis of endometriosis. Numerous immunological abnormalities have been described, including increased production of proinflammatory cytokines/chemokines and alterations in B cell activation and T/B cell function [3,4,5,6,7,8]. Macrophages and, in particular, peritoneal macrophages, seem to play a pivotal role in the pathogenesis of endometriosis [9]. These cells play a critical role in the regulation of the inflammatory response and react to changes in the microenvironment, acquiring different phenotypes. By similarity to Th1 and Th2 responses, macrophages have been classified as M1 when involved in the induction of inflammation and M2 if they have anti-inflammatory and proliferative functions. It is important to note that this classification oversimplifies the wide range of phenotypes to adapt to different situations and that M1 and M2 only represent two extreme situations [10,11]. In the case of endometriosis, some data suggest a dysregulation of the macrophages, contributing to the disease establishment and progression [12,13,14].

Among the factors involved in this alteration in macrophage activity in endometriosis, Small Extracellular Vesicles (sEVs) have been described to play a role in favoring the switch to a specific phenotype [15]. sEVs are small membranous vesicles secreted by almost all cell types and have emerged as critical mediators of cell–cell communication and signaling in both physiological and pathological contexts. They carry a cargo of bioactive molecules, including proteins, lipids, and nucleic acids, which can be transferred between cells to modulate recipient cell behavior [16]. Over recent years, several studies have focused on the role of sEVs in mediating pathophysiological processes associated with endometriosis [17]. Among other effects, it has been suggested that sEVs may be involved in inducing a switch to an M2 phenotype in macrophages. However, the changes reported are generally limited and inconsistent among the different studies available, in part due to variations in exosome sources (menstrual blood, peritoneal lavage, endometrioma biopsies) and experimental models.

The aim of our study is to characterize the potential effect of circulating sEVs in the plasma of well-characterized patients with endometriosis on macrophage phenotype.

## 2. Methods

A single-center case-control study was conducted. All the participants were prospectively and consecutively recruited during the same 6-month period from September 2022 to February 2023. The study was approved by the local Ethical Committee according to prevailing regulations (HCB/2020/1445). Written informed consent was obtained from all participants.

### 2.1. Patients

Two groups of patients were recruited. The endometriosis group (E group) consisted of patients diagnosed with endometriosis by imaging testing (gynecological ultrasonography and/or magnetic resonance imaging), confirmed by histopathologic study (n = 12). Endometriosis patients underwent surgery due to painful symptoms and/or infertility. The control group (C group) included women who underwent laparoscopy for tubal sterilization without presurgical suspicion of endometriosis and without endometriosis or signs of any inflammatory pelvic condition during the surgical procedure (n = 12).

The inclusion criteria included women aged 18–40 years with a body mass index (BMI) of less than 30.00 kg/m^2^. The exclusion criteria encompassed a history of past or present malignancy, endocrine, cardiovascular, or systemic diseases, pregnancy or breastfeeding within 6 months prior to sample collection, premature ovarian failure or menopausal status, use of hormonal contraception or other hormonal treatments such as GnRH analogs within 6 months prior to sample collection, or having had an inflammatory or infectious condition within 6 months prior to sample collection. Clinical and epidemiological data were collected from all participants, including age, BMI, smoking status, and parity. Before surgery, patients were asked to rate dysmenorrhea, dyspareunia, and chronic pelvic pain on a 0- to 10-point numerical rating scale (NRS), with “0” indicating no pain and “10” indicating the worst possible pain. Laparoscopy was performed in all patients as described elsewhere [18]. The pelvic organs and peritoneum were examined before proceeding with the indicated surgical intervention in each case. All removed tissue was sent for pathological analysis to either confirm or rule out endometriosis. After undergoing laparoscopy and histological examination, patients were conclusively categorized into one of the two patient groups.

### 2.2. Sample Collection

Venous blood samples were taken at the time of recruitment through an antecubital venous puncture prior to the administration of pre-anesthetic medication and before the induction of anesthesia. The samples were collected in tubes containing 3.8% trisodium citrate (1:9, *v*:*v*) (Becton Dickinson, Rutherford, NJ, USA). Platelet-free plasma was immediately prepared by double centrifugation at 2000× *g* for 10 min at 22 °C, followed by 5000× *g* for 10 min at 4 °C. The plasma was then aliquoted and stored at −80 °C.

### 2.3. sEV Isolation

sEVs were isolated as previously described [19], with some modifications. Plasma samples (1 mL) were centrifuged at 2000× *g* and 10,000× *g* for 10 and 30 min, respectively, at 4 °C. Then, the supernatant was recovered, resuspended in 28 mL of phosphate-buffered saline (PBS), filtered through a 0.22 µm filter, and ultracentrifuged at 120,000× *g* for 120 min. After that, the pelleted vesicles were washed with PBS and centrifuged again at 120,000× *g*. The quality of preparation was verified by nanoparticle tracking analysis and by determining the presence of the sECv markers, Alix, and CD9 and the absence of calnexin CNX by Western blot. The number of sEVs obtained was also quantified by measuring their protein content using the bicinchoninic acid protein assay kit (Pierce, Rockford, IL, USA).

### 2.4. Nanoparticle Tracking Analysis

The size distribution and concentration of sEVs were determined using a NanoSight LM10 device (NanoSight, Salisbury, UK). Consistent parameters were applied to all samples, and three 1-min videos were recorded for each. Background levels were assessed using filtered PBS, which showed no signal.

### 2.5. SDS-PAGE and Western Blot

Proteins from sEVs were extracted using RIPA buffer (10 mM Tris pH 8.0, 140 mM NaCl, 1% Triton X-100, 1 mM EDTA, and 0.1% SDS) with added protease inhibitors. The protein concentration was measured with a Bradford assay. Ten micrograms of protein were separated on a 12% SDS-PAGE gel and then transferred to a polyvinylidene difluoride (PVDF) membrane (Immun-Blot, Bio Rad, CA, USA) using wet transfer conditions. The PVDF membranes were blocked for one hour in 5% nonfat milk in PBS, followed by an overnight incubation at 4 °C with antibodies against CD9 (2059782-1-AP; 1:1000), ALIX (12422-1-AP; 1:1000), and calnexin (10427-2-AP; 1:1000) (ProteinTech, Sankt Leon-Rot, Germany). The Western blots were then washed and incubated for 1 h and 30 min at room temperature with a DyLight 800-conjugated secondary antibody (1:10,000) (Thermo Scientific, Waltham, MA, USA). Immunoreactive bands were visualized using the Odyssey Infrared Imaging System, Image Lab Touch Software version 2.3.0.07 (LI-COR Biosciences, Lincoln, NE, USA).

### 2.6. Cell Culture

Human THP-1 cells were cultured in suspension in RPMI 1640 medium supplemented with 10% fetal bovine serum (FBS; Gibco^TM^, Thermo Fisher Scientific), 2 mM L-glutamine, 100 U/mL penicillin, and 100 µg/mL streptomycin in 24-well plates (200,000 cells/well). Cells were differentiated into macrophages through a first incubation with 100 nM phorbol 12-myristate 13-acetate (PMA) for 24 h. After that, the media containing PMA was discarded and replaced with fresh media without PMA for a further 24 h. To evaluate the effect of sEVs on macrophages, cells were incubated with sEVs for 24 h, and the changes in the expression of inflammatory cytokines or phenotype markers were evaluated by duplicate by reverse transcriptase-polymerase chain reaction (RT-PCR). To assess the effect sEVs had on activated macrophages, in some experiments, these were pretreated one hour earlier with lipopolysaccharide (LPS) (100 ng/mL), which is an inflammatory stimulus. In some experiments, cells were also treated with 1 µM rosiglitzone, a PPARG agonist, for 6 h.

### 2.7. RNA Isolation and qPCR

Total RNA from cells was extracted using the TRizol reagent (Invitrogen, CA, USA). RNA was quantified by measuring the absorbance at 260 and 280 nm using a NanoDrop ND-1000 spectrophotometer (NanoDrop Technologies, DE, USA). cDNA was synthesized from a 1 µg RNA sample using the iScript cDNA synthesis kit (Bio-Rad, Hercules, CA, USA).

Subsequent quantitative PCR (qPCR) was performed in a DNA Engine, Peltier Thermal Cycler (Bio-Rad, Hercules, CA, USA) using iTaq^TM^ Universal SYBR^®^ Green Supermix (Bio-Rad, Hercules, CA, USA) and the corresponding primers (Table 1). Reactions were performed in duplicate and threshold cycle values were normalized to *GAPDH* gene expression. The specificity of the products was determined by melting curve analysis. The relative expression of target genes to *GAPDH* was calculated by the ΔC(t) formula.

### 2.8. sEVs and Cells Staining

For internalization experiments, sEVs were tagged with PKH26 red fluorescent cell linker dye (Sigma-Aldrich, St. Louis, MO, USA) for 3 min. The staining process was halted by adding 3% bovine serum albumin (BSA) for 1 min. To eliminate any unbound dye, sEVs underwent three washes with PBS using 300 kDa Nanosep centrifugal devices (Pall Corporation, New York, NY, USA). THP-1 macrophages were labeled with PKH67 green, a fluorescent cell linker dye for the overall cell membrane. Images were obtained using an inverted Olympus CFX53 fluorescence microscope and the Cellsens standard 1.18 software.

### 2.9. Confocal Microscopy

Cells were also observed using an inverted Nikon Eclipse Ti2-E microscope (Nikon Instruments, Tokyo, Japan) attached to the spinning disk unit Andor Dragonfly. Samples were excited with 405 nm and 488 nm laser diodes. Cells were imaged on a high-resolution scientific complementary metal oxide semiconductor (sCMOS) camera (Zyla 4.2, 2.0 Andor, Oxford Instruments Company, Abingdon, UK). Fusion software version 2.4.0.14 (Andor, Oxford Instruments Company) was used for acquisition. Image processing and analysis were performed with Image J/Fiji software using ImageJ version 1.51 J (NIH, Bethesda, MD, USA; https://imagej.net/ij/ (accessed on 15 June 2024)).

### 2.10. Statistical Analysis

The sample size was decided arbitrarily, based on previous studies analyzing macrophages in endometriosis [12,14]. The statistical analysis was performed with Graphpad Prism 4.02 software. Data are presented as mean ± standard deviation or number (percentage). A Shapiro–Wilk test was used to ascertain whether continuous variables had a normal distribution. Data were analyzed using a two-tailed Student’s *t*-test for comparison of two groups and a one-way analysis of variance (ANOVA) analysis followed by Tukey’s post-test when comparing three groups when non-treated cells were included. Statistical significance was considered when *p* < 0.05.

## 3. Results

### 3.1. Patient Characteristics

The mean age, BMI, and tobacco use were similar in the two groups analyzed. As expected, the mean NRS pain score and infertility were higher in the E group (Table 2).

Among the 12 patients included in the E group, the following endometriosis types and locations were recorded: uterosacral ligaments (n = 11), torus uterinus (n = 8), rectosigmoid (n = 3), vesical (n = 2), ureteral (n = 1), other intestinal location (n = 1), and vaginal (n = 1). All endometriosis implants were excised during surgery. Ovarian endometriomas were found in 7 patients (84%), and superficial peritoneal endometriosis was identified in 9 patients (72%).

### 3.2. sEVs Characterization

The size of the purified particles measured by the nanoparticle tracking analysis agreed with that expected for sEVs (<200 nm), although small amounts of other populations with larger sizes could be detected in both groups [17]. No differences were detected in the size of the sEVs obtained from the plasma of women in both the E and C groups (Figure 1A). Western blot analysis confirmed the presence of the sEVs markers CD9 and Alix, as well as the absence of the negative marker calnexin (Figure 1B) [20].

### 3.3. Macrophage Exosome Uptake

Red PKH26 labeled exosome uptake by THP1 macrophages was observed under a fluorescent microscope after 2 h and 4 h of incubation. At two hours, many macrophages had incorporated sEVs, and the vast majority had uptaked the sEVs at 4 h (Figure 2A). However, there were no significant differences in either the 2 h or 4 h evaluation in terms of the number of sEVs internalized by macrophages in women from the C and E groups (Figure 2B). High-magnification confocal microscopy analysis (Figure 2C) confirmed that sEVs had indeed been taken up by cells in both groups.

### 3.4. Changes in Macrophage Gene Expression and Phenotype

RT-PCR analysis of gene expression revealed that treatment with endometriosis-derived sEVs resulted in a small, albeit significant, reduction of interleukin-1 beta (IL1B) expression and a higher expression of MRC1 in macrophages, which, in this case, did not reach significance (Figure 3A). Both changes are characteristics of a switch to an M2 phenotype. In the case of PPARG, macrophages treated with sEVs from endometriosis also showed higher expression levels than those treated with sEVs from controls, although in this case, it was not accompanied by any change in the expression of CD36, one of the genes commonly induced by PPARG activation.

The effect of sEVs was also measured in LPS-activated macrophages. As expected, under this proinflammatory stimulus, IL1B expression in macrophages was greatly increased, while MRC1 was strongly inhibited. Treatment with control sEVs did not alter these changes, whereas endometriosis sEVs reduced IL1B expression and did not generate significant changes in MRC1 (Figure 3B).

### 3.5. Effect of PPARG Activation

With PPARG expression being induced by endometriosis sEVs, we aimed to assess the effect of pharmacological activation of PPARG on CD36 and the macrophage phenotype. For this purpose, cells were treated with rosiglitazone, a PPARG agonist. Activation with rosiglitazone resulted in an increase in CD36 expression in macrophages in both control and endometriosis exosome-treated cells (Figure 4). However, rosiglitazone did not modify the changes in macrophage phenotype induced by endometriosis sEVs, with no significant changes in IL1B or MRC1 expression being observed.

## 4. Discussion

It is well-known that macrophages play a determining role in the pathogenesis of endometriosis. In different experimental approaches, it has been described that macrophages and, particularly, peritoneal macrophages, tend to acquire an M2 phenotype in endometriosis patients. This allows macrophages to ectopically promote adhesion, proliferation, or angiogenesis of endometrial stromal cells. There are also data suggesting that sEVs may play a prominent role in the process of changing the phenotype. In this case, the evidence is less consistent due to the variety of experimental approaches.

The effect on macrophages seems clear when using sEVs derived from cultured endometriosis cells or in experimental animal models. In the case of plasma, the effect on the change of phenotype has also been described, but the interest has mainly focused on the use of the content of sEVs as biomarkers. In this sense, different studies analyzing both miRNAs [21] and lncRNAs [22] have focused on the utility of sEVs as biomarkers and point out different pathways that may be involved in their effects, including Smad2/Smad3 or PTEN-PI3K [12,21].

In relation to sEVs obtained from the plasma of patients, their effects are more subtle, as the amount of sEVs originating from the endometrial tissue constitutes a very small fraction of the total circulating sEVs. Presumably, for this reason, the impact on the macrophage phenotype observed in our study was limited. While the tendency toward an M2 phenotype, characterized by a lower expression of IL1B and TNF and a trend to an increase in MRC1 and ARG1, aligns with findings from other studies using sEVs derived from endometrial tissue or endometrial cells, our study revealed modest changes. The same pattern emerged when we tested macrophages treated with an inflammatory stimulus with LPS. The response to LPS, marked by a substantial increase in IL1B expression, was attenuated when macrophages had been treated with endometriosis sEVs, but the magnitude of this reduction was moderate. It should be noted that in this study we used THP1 cells differentiated into macrophages. Although these cells reflect the vast majority of characteristics of macrophages and are widely used in these types of studies, they are known to readily shift into M1, but are more difficult to shift into M2 phenotype.

Neither did we detect any change in the levels of exosome uptake. The level and rate of incorporation of both control and endometriosis sEVs by THP1 macrophages were similar. This rules out that the observed effects are simply the result of higher uptake in the case of endometriosis. Some studies have described a higher uptake of sEVs obtained from endometrial tissue compared to those from control plasma, although the fact of using exosomes from different origins complicates comparisons [22]. In our case, both types of sEVs were obtained from plasma, and again, the dilution of sEVs from endometriotic tissue in total plasma may mask potential changes in uptake.

An interesting finding was the induction of PPARG in macrophages treated with endometriosis sEVs. PPARG agonist treatment was proposed for endometriosis as it appeared to prevent or reduce some of its features [23,24]. In our model, the induction of PPARG by sEVs did not seem to be directly linked to its activation, as evidenced by the absence of changes in the expression of CD36, a scavenger receptor known to be upregulated by PPARG [24]. Consequently, we opted to pharmacologically activate PPARG by treating macrophages with rosiglitazone to investigate whether this PPARG agonist could reverse the shift to the M2 phenotype promoted by endometriosis sEVs. Regrettably, although rosiglitazone did induce the expression of CD36, no discernible changes were observed in the levels of IL1B or MRC1 expression. This finding suggests that the potential effect described by PPARG agonists may not be directly related to the influence of sEVs on the macrophage phenotype.

Our study has several strengths. This prospective case-control study involved all patients undergoing surgery. Based on surgical findings and histological analysis, patients were definitively assigned to either the study group or the control group rather than being classified solely based on the presurgical evaluation. Therefore, the healthy controls were found to have no endometriosis lesions. Patients with other disorders were excluded from our study. Finally, all blood samples were collected immediately prior to surgery to evaluate the baseline profile of the patients.

However, as a limitation, it must be stressed that the sample size was small and arbitrarily decided based on previous studies investigating macrophage activation in endometriosis; therefore, one must be cautious when generalizing them. Finally, it is worth noting that in future studies, it would be interesting to compare sEVs derived from plasma with those from other sources, such as peritoneal or endometrial samples.

In conclusion, we found that circulating sEVs in women with endometriosis show a certain capacity to shift the activation state of macrophages toward an M2 phenotype, but this shift is limited and does not modify the level of uptake or the response to PPARG ligands.

## Figures and Tables

**Figure 1 biomolecules-14-00737-f001:**
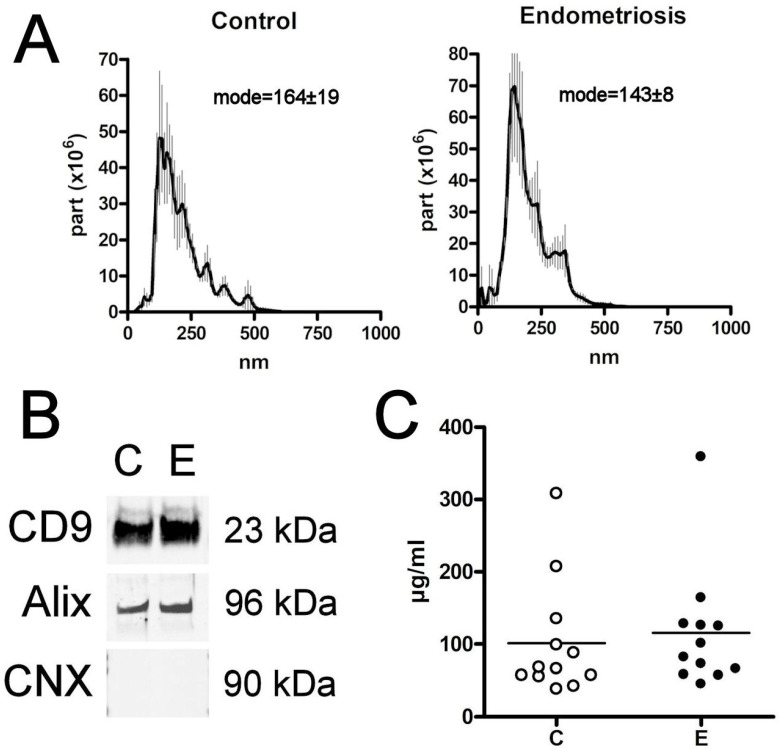
(**A**) Nanovesicle tracking assay confirms that the size of extracellular vesicles obtained corresponds to sEVs. (**B**) Western blot analysis of sEVs proteins CD9 and Alix from the different groups. Calnexin (CNX) was included as a negative control (the original is included in Appendix A). (**C**) EV levels, expressed as μg prot/mL, detected in the plasma. C = Control; E = Endometriosis.

**Figure 2 biomolecules-14-00737-f002:**
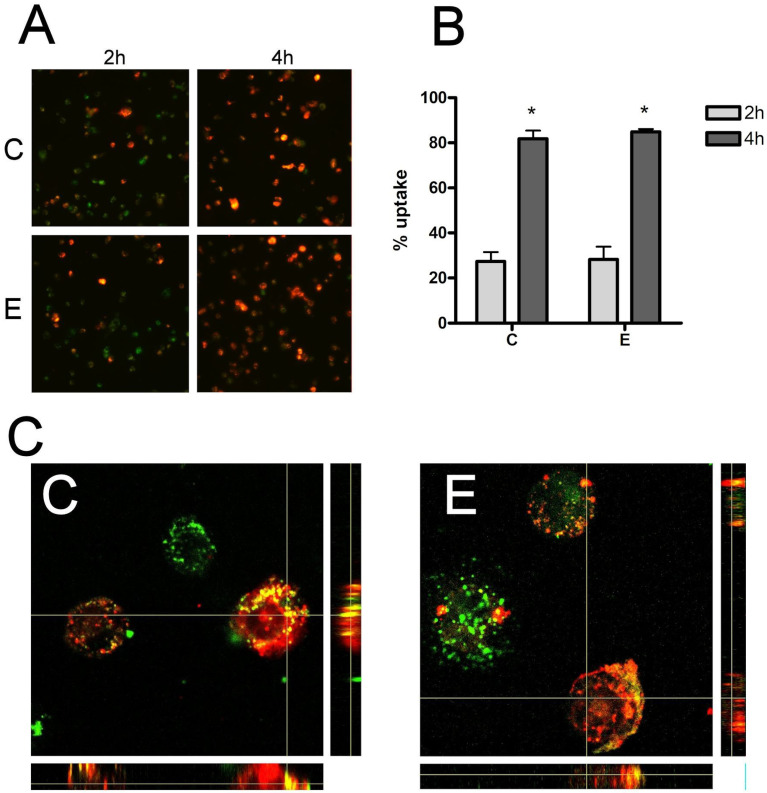
(**A**) Cell uptake of sEV by THP1 macrophages 2 h and 4 h (10×). (**B**) No relevant changes were observed in sEVs uptake by THP1 macrophages at 2 h or 4 h. (**C**) Confocal analysis of sEVs uptake (60×). sEVs were stained with red PKH26 dye and cells were stained with green PKH67 dye. C = Control E = Endometriosis. * = *p* < 0.05 vs. 2 h.

**Figure 3 biomolecules-14-00737-f003:**
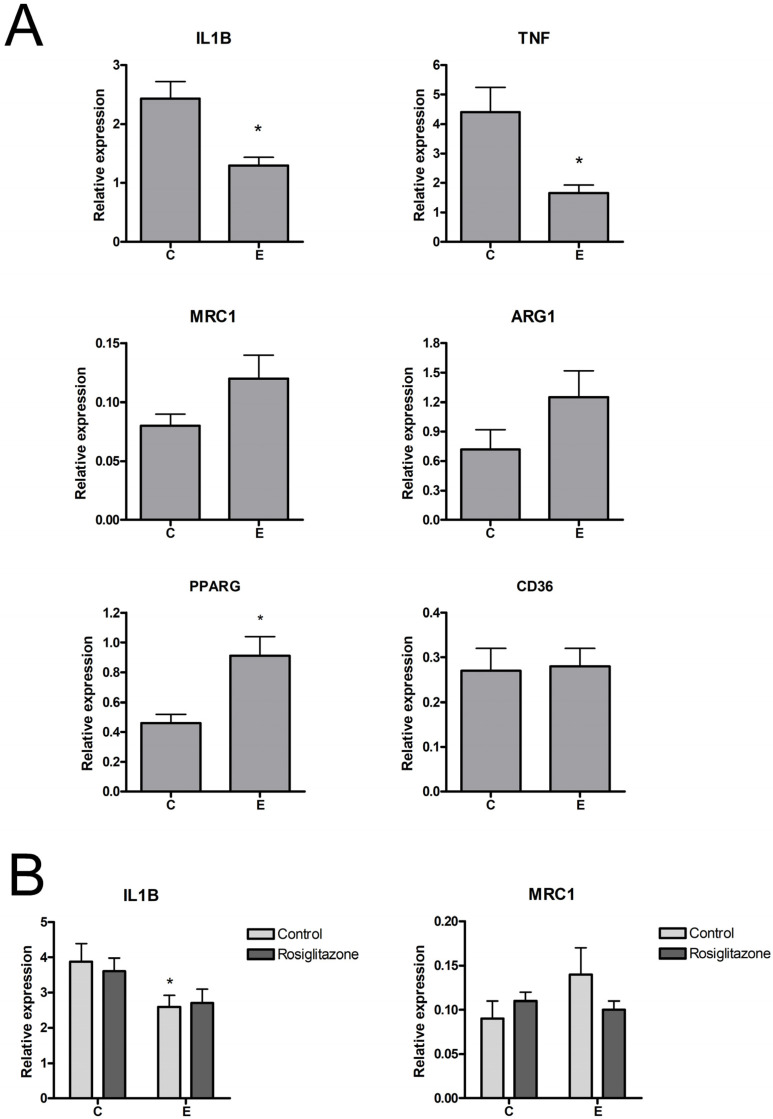
(**A**) Induction of IL1B, TNF, MRC1, ARG1, PPARG, and CD36 expression in THP1 macrophages treated with 10 µg/mL sEVs for 24 h. (**B**) Effect of sEVs on the expression of IL1B and MRC1 in THP1 macrophages treated with LPS (100 ng/mL). C = Control; E = Endometriosis. * = *p* < 0.05 vs. C.

**Figure 4 biomolecules-14-00737-f004:**
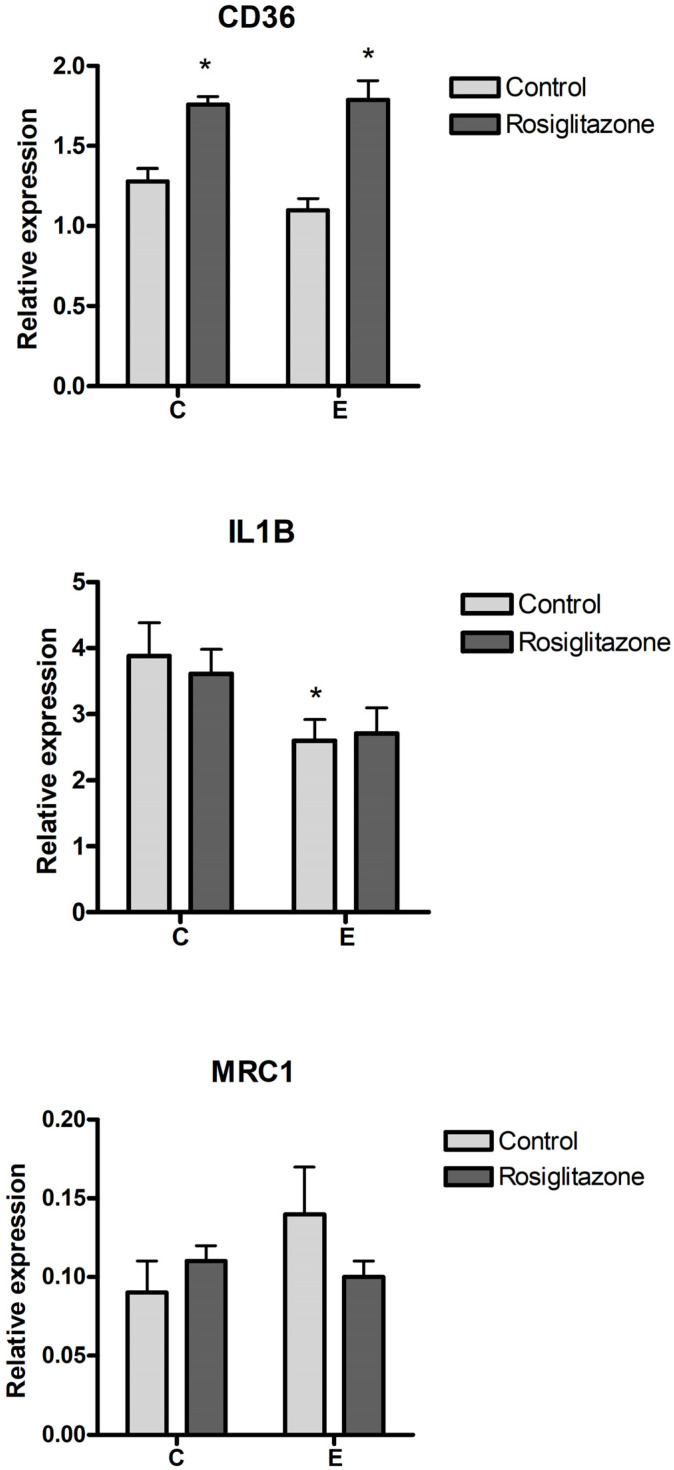
Effect of pharmacological activation of PPARG on CD36 and macrophage phenotype using rosiglitazone treatment. C = Control; E = Endometriosis. * = *p* < 0.05 vs. C.

**Table 1 biomolecules-14-00737-t001:** Primers used for quantitative polymerase chain reaction.

Gene	Accession Number	Product Size	Sequence
*GAPDH*	NM 002046.3	97 bp	Forward: 5′-GATCATCAGCAATGCCTCCT-3′ Reverse: 5′-TGTGGTCATGAGTCCTTCCA-3′
*IL1B*	NM 000576.2	120 bp	Forward: 5′-GGACAAGCTGAGGAAGATGC-3′ Reverse: 5′-TCGTTATCCCATGTGTCGAA-3′
*TNF*	NM 000594.4	114 bp	Forward: 5′-AGCCCATGTTGTAGCAAACC-3′ Reverse: 5′-GGCACCACCAACTGGTTATC-3′
*MRC1*	NM 002438	131 bp	Forward: 5′-GGATGGATGGCTCTGGTG-3′ Reverse: 5′-TCTGGTAGGAAACGCTGGT-3′
*ARG1*	NM 000045.3	160 bp	Forward: 5′-ACACTCCATTGACAACCACA-3′ Reverse: 5′-TCCACGTCTCTCAAGCCAAT-3′
*PPARG*	NM 138712.3	208 bp	Forward: 5′-TTGCAGTGGGGATGTCTCAT-3′ Reverse: 5′-TTTCCTGTCAAGATCGCCCT-3′
*CD36*	NM 001127444.1	150 bp	Forward: 5′-AGATGCAGCCTCATTTCCAC-3′ Reverse: 5′-GCCTTGGATGGAAGAACAAA-3′

**Table 2 biomolecules-14-00737-t002:** Demographic and clinical data of the two groups. NRS: numerical rating scale; BMI: body mass index. Data are provided as mean ± standard deviation or number (percentage).

	Control Group (n = 12)	Endometriosis Group (n = 12)	*p* Value
Age (years)	33.1 ± 3.6	34.2 ± 4.1	0.14
BMI (Kg/m^2^)	24.5 ± 4.6	24.8 ± 3.9	0.56
Tobacco use	3 (25)	4 (33.3)	0.21
Nulliparous	5 (41.6)	6 (50)	0.30
Dysmenorrhea (NRS)	0.7 ± 2.3	8.7 ± 3.2	<0.001
Chronic pelvic pain (NRS)	0.0 ± 0.0	7.4 ± 2.3	<0.001
Dyspareunia (NRS)	0.0 ± 0.0	6.5 ± 2.4	<0.001

## Data Availability

Data are contained within the article and Appendix A.

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
