# Peer review of "Macrophage Phenotype Induced by Circulating Small Extracellular Vesicles from Women with Endometriosis"

_biomolecules, 2024, doi:10.3390/biom14070737_

Round 1
Reviewer 1 Report
Comments and Suggestions for Authors
In this study, the authors characterized the potential effect of circulating exosomes in the plasma of patients with endometriosis on macrophage phenotype. However, below are several details that require further precision or correction before the work can be published:
1. Methods, patients: how was n calculated?
2. Lane 91: Table I does not exist. Replace it with Table 1. Check this throughout the ms.
3. Lane 96: Replace "indexx" with "index".
4. Table 1: Correct the format. Some data is displaced.
5. Lane 111: Replace "described[19]" with "described [19]".
6. Lane 112: Replace "ml" with "mL". Use the international system of units. Check this throughout the ms.
7. Lanes 113-114: Correct fon color.
8. Lane 131: Correct font color.
9. Lane 142: Replace "ml" with "mL". Use the international system of units.
10. Lane 145: Replace "2h" with "2 h". Check this throughout the ms.
11. Methods, cell culture: the authors should describe the experiment in more detail. How many cells did they seed? in which plate did they seed them? What was the n? How many replicates?
12. Lane 155: the correct name is iTaqTM Universal SYBR® Green Supermix (Bio-Rad, CA).
13. Table 2: Correct the format. Some data is displaced.
14. Table 2: To improve quality, please add the gene accession number and the amplified size (bp).
15. Methods, statistical analysis: What was the software version used? Did all the data have a normal distribution? Describe the non-probabilistic sampling used. Was it convenience sampling?
16. Methods, statistical analysis: Reference 20, on which the sampling is based, used 28 women in a single group. I have concerns about the statistical power of the study, given that there are 12 women per study group.
17. Methods, statistical analysis: The study suggests that there are two groups (control and endometriosis). However, they then talk about three groups. More precision is necessary.
18. Table 1, this table is a result and therefore should appear in the results section and not in the methods.
19. Lanes 180-194: For easy reading, the authors might consider summarizing this data in a table.
20. Lanes 186-187: How much was expected for exosomes. You could support this finding with a reference.
21. Figure 1A: The quality and presentation of the graphics must be improved.
22. Figure 1B: In the control group, a trace for the negative control can be seen. The original uncropped images of the blots are not included. The quality of the blots must be improved. Apparently, in the endometriosis group there are more exosomes, why?
23. Lane 197: Replace "two hours" with "2 h". Use the international system of units. Check this throughout the ms.
24. Lane 198: "4 hours" with "4 h". Use the international system of units. Check this throughout the ms.
25. Replace "2h or 4h" with "2 h or 4 h". Use the international system of units. Check this throughout the ms.
26. Figure 2: "No relevant changes were observed in exosome uptake by THP1 macrophages at 2h or 4h". This conclusion is not obvious. Please add the quantification of this result. In addition, replace "2h or 4h" with "2 h or 4 h". Use the international system of units. Check this throughout the ms.
27. Lane 213: Why were macrophages treated with LPS? What is the rationale? This does not appear in the methods. This experiment should be detailed in the methods section.
28. Lane 214 and 216: In methods this gene (IL1B) appears as IL-1B. Use Human Gene annotation. Check this throughout the ms.
29. Lane 215 and 217: Check Human Gene annotation for MRC1.
30. Figure 3: The quality of the graphics should be improved. Replace "ml" with "mL". Replace "24h" with "24 h". Check Human Gene annotation.
31. Lanes 224-226: PPAR is induced in control group too (Figure 3A). More care must be taken with the idea that is proposed.
32. Figure 4. Check Human Gene annotation. Replace "- Effect of pharmacological..." with "Effect of pharmacological...".
Author Response
- Methods, patients: how was n calculated?
Since the study was focused on clarify lack of congruence between previous studies on endometriosis, we have selected an arbitrary number of patients, based on previous studies on exosomes. It has been described in the “Statistical analysis” paragraph. For the in vitro studies, n=12 usually is enough to obtain reliable conclusions. This fact was already stressed in the limitations of the study at the end of the Discussion.
- Lane 91: Table I does not exist. Replace it with Table 1. Check this throughout the ms.
Table I has been replaced by Table 1.
- Lane 96: Replace "indexx" with "index".
Index has been corrected.
- Table 1: Correct the format. Some data is displaced.
Format has been fixed.
- Lane 111: Replace "described[19]" with "described [19]".
This, and the rest of references, has been fixed.
- Lane 112: Replace "ml" with "mL". Use the international system of units. Check this throughout the ms.
ml has been replaced by mL all around the text.
- Lanes 113-114: Correct fon color.
Color has been corrected.
- Lane 131: Correct font color.
Color has been corrected.
- Lane 142: Replace "ml" with "mL". Use the international system of units.
mL has been fixed.
- Lane 145: Replace "2h" with "2 h". Check this throughout the ms.
24 h has been corrected in all places.
- Methods, cell culture: the authors should describe the experiment in more detail. How many cells did they seed? in which plate did they seed them? What was the n? How many replicates?
As suggested, methods of cell culture have been explained in more detail.
- Lane 155: the correct name is iTaqTM Universal SYBR® Green Supermix (Bio-Rad, CA).
Name has been corrected.
- Table 2: Correct the format. Some data is displaced.
Table 2 has been fixed
- Table 2: To improve quality, please add the gene accession number and the amplified size (bp).
Accession number and product size has been included.
- Methods, statistical analysis: What was the software version used? Did all the data have a normal distribution? Describe the non-probabilistic sampling used. Was it convenience sampling?
Software version has been included. Normal distribution was assessed and this information is now included in the revised manuscript in the Statistical analysis section. And, as previously indicated in methods “All the participants were prospectively and consecutively recruited along the same 6-month period”
- Methods, statistical analysis: Reference 20, on which the sampling is based, used 28 women in a single group. I have concerns about the statistical power of the study, given that there are 12 women per study group.
As we pointed previously, we used a small number of patients since we are focused on clarify incongruences on previous studies (please see answer to your concern number 1).
- Methods, statistical analysis: The study suggests that there are two groups (control and endometriosis). However, they then talk about three groups. More precision is necessary.
In some experiments, the untreated cells are also included. Clarification has been included in the Statistical analysis Section.
- Table 1, this table is a result and therefore should appear in the results section and not in the methods.
As suggested, Table 1 has been moved to results section (please note that now is Table 2)
- Lanes 180-194: For easy reading, the authors might consider summarizing this data in a table.
Since this information refers only to endometriosis patients, we consider that a table with only one column in not useful. We have tried to maintain a low number of tables in the paper.
- Lanes 186-187: How much was expected for exosomes. You could support this finding with a reference.
In results it now is indicated that for sEVs we expected a size <200 nm although small amounts of other populations with larger sizes could be detected in both groups. A reference has been included to support our contention.
- Figure 1A: The quality and presentation of the graphics must be improved.
Figure 1 has been completely modified and improved.
- Figure 1B: In the control group, a trace for the negative control can be seen. The original uncropped images of the blots are not included. The quality of the blots must be improved. Apparently, in the endometriosis group there are more exosomes, why?
Original figure was replaced by a new western blot, with better quality and the additional markers TSG101, Alix and CD9 instead of CD63.
- Lane 197: Replace "two hours" with "2 h". Use the international system of units. Check this throughout the ms.
All suggested changes have been made
- Lane 198: "4 hours" with "4 h". Use the international system of units. Check this throughout the ms.
All suggested changes have been made
- Replace "2h or 4h" with "2 h or 4 h". Use the international system of units. Check this throughout the ms.
All suggested changes have been made
- Figure 2: "No relevant changes were observed in exosome uptake by THP1 macrophages at 2h or 4h". This conclusion is not obvious. Please add the quantification of this result. In addition, replace "2h or 4h" with "2 h or 4 h". Use the international system of units. Check this throughout the ms.
As suggested, a graph with the quantification has been added and typing errors have been corrected.
- Lane 213: Why were macrophages treated with LPS? What is the rationale? This does not appear in the methods. This experiment should be detailed in the methods section.
In methods we added this information: “To assess the effect sEVs had on activated macrophages, in some experiments these were pretreated one hour earlier with lipopolysaccharides (LPS) (100 ng/mL), which is an inflammatory stimulus”
- Lane 214 and 216: In methods this gene (IL1B) appears as IL-1B. Use Human Gene annotation. Check this throughout the ms.
Human gene annotation has been used all around the ms.
- Lane 215 and 217: Check Human Gene annotation for MRC1.
Human gene annotation has been used all around the ms.
- Figure 3: The quality of the graphics should be improved. Replace "ml" with "mL". Replace "24h" with "24 h". Check Human Gene annotation.
All suggested changes have been included.
- Lanes 224-226: PPAR is induced in control group too (Figure 3A). More care must be taken with the idea that is proposed.
Sentence has been rewritten: “In the case of PPARG, macrophages treated with sEVs from endometriosis also showed higher expression levels than those treated with sEVs from controls”
- Figure 4. Check Human Gene annotation. Replace "- Effect of pharmacological..." with "Effect of pharmacological...".
Suggested change has been included.
Reviewer 2 Report
Comments and Suggestions for Authors
Answer to Scientific Article evaluation Opinion
Title: The Role of Extracellular Vesicles as Modulators of the Tumor Microenvironment
Date: 05/17/2024
General Comment. The article presented is very interesting and of great value to the scientific and medical community for, for example, diagnosis and/or prognosis of endometriosis through a non-invasive method. Although the richness in the applied experiments and the correlations between the biomarkers chosen to achieve the objective of the study, there are some divergences to be explained and/or experiments to be added to justify such divergences. Below are my notes and questions.
Notes and Questions:
#01 – Why is the introduction in the summary so small? I believe that at least two lines of introduction enrich the article and call for the search as a reference.
#02 – I didn’t find the ethics committee approval process number.
#03 – Very small and non-homogeneous sample size to justify statistically non-significant results and/or trends.
#04 - Table 1, in addition to being disconfigured, I believe is missing data relating to the NRS.
#05 - Why use anesthetic during collection? Can't it influence cellular response?
#06 - Are there no extracellular microvesicles between 150 nm and 220 nm that could have been retained in the 0.22 µm filter with the exosomes? If not, justify.
#07 – According to the protocol for anti-CD63 antibody (25682-1-AP), which was used in this study, it involves the use of 20 µg of protein and incubation at 37ºC for 1h. Why was only 10 µg used and incubation at 4ºC overnight?
#08 – To confirm the presence of only exosomes in the samples, I consider labeling with CD9 or CD81 for greater reliability in the results. Even showing labeling with calnexin (negative marker for exosomes).
#09 - According to the protocol for Calnexin (10427-2-AP), which was used in this study, it is only used in cells (10 µg) and diluted 1:20000. In the present study, plasma and 20 µg were used at a dilution of 1:10000. Was any test carried out beforehand to confirm the dilution carried out? Or, was any reference followed to apply the presented protocol?
#10 – How were the statistical tests chosen? Knowing that the sample n is small, I believe that using tests based on Normality (Normal or Gaussian curve) - normal distribution when all values found are close or very close to the average value is of great value. This is because, if the data comes from two normal populations with the same variances, the classic 2-sample t-test is as or more powerful than the Welch t-test. The assumption of normality is not critical to the classical procedure (Pearson, 1931; Barlett, 1935; Geary, 1947), but the assumption of equality of variances is important to ensure valid results. More specifically, the classical procedure is sensitive to the assumption of equality of equal variances when sample sizes differ regardless of how large the samples are (Welch, 1937; Horsnell, 1953). In practice, however, the assumption of equality of variances is rarely true, which can lead to higher error rates in the classic t-test. Therefore, if the classic 2-sample t-test is used when two samples have different variances, the test is more likely to produce incorrect results. The Welch t-test is a viable alternative to the classical t-test because it does not assume equality of variances and is therefore insensitive to different variances for all sample sizes. However, Welch's t-test is based on an approximation of values (similarity between variances).
I believe that, due to the simple fact of more effective targeting in statistical analysis, the results have more real values for the small sample size presented.
#11 – The figures presented in the results have low resolution which, I believe, greatly hampers confirmation and/or questions. I suggest increasing the resolution of the images.
#12 – The discussion needs to be further worked on given the limitations of the study and/or add the experiments suggested in #08 or increase the sample size. After this, I believe that the discussion will be enriched and better elaborated with fewer limitations.
Author Response
#01 – Why is the introduction in the summary so small? I believe that at least two lines of introduction enrich the article and call for the search as a reference.
We agree with you and, as suggested, we have included a longer introduction in the revised manuscript.
#02 – I didn’t find the ethics committee approval process number.
We are sorry for this mistake. It has been included in the revised manuscript.
#03 – Very small and non-homogeneous sample size to justify statistically non-significant results and/or trends.
It has been stressed as a limitation in the Discussion section the following: “However, as a limitation, it must be stressed that the sample size was small and arbitrarily decided based on previous studies investigating macrophage activation in endometriosis.”
#04 - Table 1, in addition to being disconfigured, I believe is missing data relating to the NRS.
Table has been fixed. NRS of the different pain types has been relocated and now is more clear.
#05 - Why use anesthetic during collection? Can't it influence cellular response?
As indicated, collection has been carried on before the anesthesia in all patients in order to avoid any interference. Since samples were obtained before anesthetic induction, no effects could be expected.
#06 - Are there no extracellular microvesicles between 150 nm and 220 nm that could have been retained in the 0.22 µm filter with the exosomes? If not, justify.
We are not sure about the question. Filters always retain some of the vesicles by adsorption mechanisms to the material and we already have that. An effect that is surely more marked in the fraction close to the fear size limit. But potential losses affect both groups equally, so comparisons are not affected.
#07 – According to the protocol for anti-CD63 antibody (25682-1-AP), which was used in this study, it involves the use of 20 µg of protein and incubation at 37ºC for 1h. Why was only 10 µg used and incubation at 4ºC overnight?
Western blot analysis has been improved. CD63 has been replaced by CD9 and Alix has been included.
#08 – To confirm the presence of only exosomes in the samples, I consider labeling with CD9 or CD81 for greater reliability in the results. Even showing labeling with calnexin (negative marker for exosomes).
As suggested, new markers have been analyzed and western blot has been improved.
#09 - According to the protocol for Calnexin (10427-2-AP), which was used in this study, it is only used in cells (10 µg) and diluted 1:20000. In the present study, plasma and 20 µg were used at a dilution of 1:10000. Was any test carried out beforehand to confirm the dilution carried out? Or, was any reference followed to apply the presented protocol?
To clarify this concern we included a new reference [20] for our western protocol used for plasma samples.
#10 – How were the statistical tests chosen? Knowing that the sample n is small, I believe that using tests based on Normality (Normal or Gaussian curve) - normal distribution when all values found are close or very close to the average value is of great value. This is because, if the data comes from two normal populations with the same variances, the classic 2-sample t-test is as or more powerful than the Welch t-test. The assumption of normality is not critical to the classical procedure (Pearson, 1931; Barlett, 1935; Geary, 1947), but the assumption of equality of variances is important to ensure valid results. More specifically, the classical procedure is sensitive to the assumption of equality of equal variances when sample sizes differ regardless of how large the samples are (Welch, 1937; Horsnell, 1953). In practice, however, the assumption of equality of variances is rarely true, which can lead to higher error rates in the classic t-test. Therefore, if the classic 2-sample t-test is used when two samples have different variances, the test is more likely to produce incorrect results. The Welch t-test is a viable alternative to the classical t-test because it does not assume equality of variances and is therefore insensitive to different variances for all sample sizes. However, Welch's t-test is based on an approximation of values (similarity between variances).
I believe that, due to the simple fact of more effective targeting in statistical analysis, the results have more real values for the small sample size presented.
We agree with you that the sample size is small and that this issued had to be clarified in the Statistical analysis section. Moreover, please note that statistical analysis was rechecked and a normal distribution was ascertained (please see new Statistical analysis section).
#11 – The figures presented in the results have low resolution which, I believe, greatly hampers confirmation and/or questions. I suggest increasing the resolution of the images.
Figures has been modified and improved.
#12 – The discussion needs to be further worked on given the limitations of the study and/or add the experiments suggested in #08 or increase the sample size. After this, I believe that the discussion will be enriched and better elaborated with fewer limitations.
We have modified the discussion, elaborating on some concepts, but maintaining the caution associated with the limited sample size.
Reviewer 3 Report
Comments and Suggestions for Authors
The MS by Martinez-Zamora is potentially interesting in his assumption, nevertheless several major points the authors must address limit my enthusiasm for their work:
1. Authors stated that “No differences were detected in the size of the exosomes obtained from the plasma of women in both the E and C groups” (line 187-88) but, looking at the fig.1A NTA analysis clearly make evident that in C groups there are at least 6 subpopulations of exos whereas in the E group there are only 3. In addition, no quantitative data are provided by authors (i.e. how many exos in C versus E group are present in the samples? Is there any statistically significant difference?). Further, even if the Western Blots are very faint, it seems that the expression of CD63 in C and E groups is different, whereas Calnexin seems barely present in sample from C group (it must be negative). How authors could explain that?
2. Authors stated that “At two hours, many macrophages had incorporated exosomes and the vast majority had phagocytosed the exosomes at 4 hours.” (lines 197-98). It seems to me that images in Fig. 2 are obtained by immunofluorescence rather than confocal microscopy, so how authors can exclude that exos are simply adhered on the surface of the cells rather than internalized. To exclude this hypothesis, authors must perform experiments by incubating cells with exos a +4 C° or, better, make a confocal microscopy analysis. Further, all the details concerning the microscopy experiment (type of microscopy, magnification, scale bar on figure) are missing.
3. As authors stated M1/M2 polarization is a more complex phenomenon than Th1/Th2 polarization then to better substantiate their hypothesis more markers should be assessed as for ex. iNOS, Arginase, MHC-II, CD163 and not only by RT-qPCR but also by flow cytometry (see for ex. Li et al., Front. Immunol. doi: 10.3389/fimmu.2021.700009). Further, as PMA-differentiated THP-1 does not fully recapitulate the features of Monocyte-derived Macrophages (MDMs), as for ex. in Kohro et al. J. Atheroscler. Thromb. doi: 10.5551/jat.11.88, authors should demonstrate that these effects are also detectable in primary MDMs.
Minor criticism:
Authors refer to vesicles as exosomes but looking at the materials and methods section the protocol they used (only ultracentrifugation without optiprep gradient) is insufficient to guarantee exos purification as well as the procedure they used to check their preparation according to ISEV nomenclature is insufficient for the definition of their vesicles as exos. I suggest them to change the definition of exos with the more recently accepted definition of small extracellular vesicles.
Author Response
- Authors stated that “No differences were detected in the size of the exosomes obtained from the plasma of women in both the E and C groups” (line 187-88) but, looking at the fig.1A NTA analysis clearly make evident that in C groups there are at least 6 subpopulations of exos whereas in the E group there are only 3. In addition, no quantitative data are provided by authors (i.e. how many exos in C versus E group are present in the samples? Is there any statistically significant difference?). Further, even if the Western Blots are very faint, it seems that the expression of CD63 in C and E groups is different, whereas Calnexin seems barely present in sample from C group (it must be negative). How authors could explain that?
No differences in the number of vesicles between groups were detected. Data (µg protein of exosomes / mL plasma) has been included in Figure 1. New Western blot has been included with an additional marker (Alix) and replacing CD63 by CD9.
- Authors stated that “At two hours, many macrophages had incorporated exosomes and the vast majority had phagocytosed the exosomes at 4 hours.” (lines 197-98). It seems to me that images in Fig. 2 are obtained by immunofluorescence rather than confocal microscopy, so how authors can exclude that exos are simply adhered on the surface of the cells rather than internalized. To exclude this hypothesis, authors must perform experiments by incubating cells with exos a +4 C° or, better, make a confocal microscopy analysis. Further, all the details concerning the microscopy experiment (type of microscopy, magnification, scale bar on figure) are missing.
As suggested, confocal analysis has been performed to confirm that vesicles are inside the cells. Information about microscopes has been included in Methods.
- As authors stated M1/M2 polarization is a more complex phenomenon than Th1/Th2 polarization then to better substantiate their hypothesis more markers should be assessed as for ex. iNOS, Arginase, MHC-II, CD163 and not only by RT-qPCR but also by flow cytometry (see for ex. Li et al., Front. Immunol. doi: 10.3389/fimmu.2021.700009). Further, as PMA-differentiated THP-1 does not fully recapitulate the features of Monocyte-derived Macrophages (MDMs), as for ex. in Kohro et al. J. Atheroscler. Thromb. doi: 10.5551/jat.11.88, authors should demonstrate that these effects are also detectable in primary MDMs.
Changes in TNF and ARG1 expression has been included in the analysis. Although it is true that it would be better to go deeper into the markers, analyzing them by cytometry or high techniques, we believe that with the changes in expression found it is already clear that there are differences in the effect of the exosomes of the different groups. Moreover, we have included a sentence in the Discussion to the fact that THP1 macrophages, although widely used in these kinds of studies, do not fully recapitulate the characteristics of macrophages.
Minor criticism:
Authors refer to vesicles as exosomes but looking at the materials and methods section the protocol they used (only ultracentrifugation without optiprep gradient) is insufficient to guarantee exos purification as well as the procedure they used to check their preparation according to ISEV nomenclature is insufficient for the definition of their vesicles as exos. I suggest them to change the definition of exos with the more recently accepted definition of small extracellular vesicles.
We agree. As indicated, nomenclature has been changed and “exosomes” has been replaced by the more correct “Small extracellular vesicles (sEVs)”
Reviewer 4 Report
Comments and Suggestions for Authors
An interesting article concerning the possibility of reprogramming the phenotype and functionality of model macrophages with circulating extracellular vesicles from healthy women and patients with various types of endometriosis. However, the work requires some changes, namely:
1) based on the presented methods for isolating and characterization of vesicles, the authors worked not with exosomes, but with small extracellular vesicles. Exosomes range in size from 30 to 150 nm. Accordingly, it is necessary to correct the terminology in the title of the article and throughout the text.
2) In the description of Figure 1, it is necessary to clarify the average size and median, as well as the standard deviation of the size of the isolated vesicles and their concentration. Figure 1A does not correspond to the pattern of exosomes and small extracellular vesicles isolated by the proposed ultracentrifugation method combined with ultrafiltration (220 nm filter). Other drawings must be submitted.
3) On page 6, in my opinion, it is erroneously written that exosomes are phagocytosed by macrophages. Even for macrophages, the predominant mechanism of internalization is endocytosis (its varieties) and macropinocytosis. Is there any evidence that in your work this was phagocytosis?
4) Fig. 2 must be supplemented with statistical data. Usually the percentage of cells and internalized vesicles per 100 cells counted in a smear is presented.
5) There are no keywords for the article. The abstract requires revision taking into account the above comments.
Author Response
1) based on the presented methods for isolating and characterization of vesicles, the authors worked not with exosomes, but with small extracellular vesicles. Exosomes range in size from 30 to 150 nm. Accordingly, it is necessary to correct the terminology in the title of the article and throughout the text.
Terminology has been modified and “exosomes” has been replaced by the most accurate “Small extracellular vesicles or sECVs”.
2) In the description of Figure 1, it is necessary to clarify the average size and median, as well as the standard deviation of the size of the isolated vesicles and their concentration. Figure 1A does not correspond to the pattern of exosomes and small extracellular vesicles isolated by the proposed ultracentrifugation method combined with ultrafiltration (220 nm filter). Other drawings must be submitted.
Figure 1 has been improved, including information about mode and concentration as well as a new western blot. In results is also indicated that, in addition to the sEVs, small amounts of other populations of vesicles with larger sizes have been detected in both groups.
3) On page 6, in my opinion, it is erroneously written that exosomes are phagocytosed by macrophages. Even for macrophages, the predominant mechanism of internalization is endocytosis (its varieties) and macropinocytosis. Is there any evidence that in your work this was phagocytosis?
You are right. The word “phagocytosis” has been replaced by “uptake”.
4) Fig. 2 must be supplemented with statistical data. Usually the percentage of cells and internalized vesicles per 100 cells counted in a smear is presented.
A graph with the % of positive cells has been included in Figure 2.
5) There are no keywords for the article. The abstract requires revision taking into account the above comments.
Keywords has been included and abstract has been modified.
Round 2
Reviewer 1 Report
Comments and Suggestions for Authors
The ms has been improved. I still have minimal concerns.
1. Table 1. The titles of each column must be added.
2. The file called "original images" does not correspond to the uncut or edited blots.
3. In limitation, the authors should add that due to the small sample size, the results cannot be generalized.
Author Response
Thank you very much for your comments on our manuscript. We have addressed your new concerns:
- Table 1. The titles of each column must be added.
We are sorry for the mistake. We have added them.
- The file called "original images" does not correspond to the uncut or edited blots.
Please note that a Supplementary Figure 1 with the Original image of Figure 1B has been added.
- In limitation, the authors should add that due to the small sample size, the results cannot be generalized.
We have added this limitation with the sentence “However, as a limitation, it must be stressed that the sample size was small and arbitrarily decided based on previous studies investigating macrophage activation in endometriosis, therefore, one must be cautious when generalizing them.”
Reviewer 2 Report
Comments and Suggestions for Authors
I am satisfied with the corrections and changes suggested, in addition to resolving my doubts. I believe this article will contribute a lot.
Author Response
Thank you very much for your nice comments on our manuscript.
Reviewer 3 Report
Comments and Suggestions for Authors
The Authors positively answered all the concerns I’ve raised then I consider the Manuscript byMartinez-Zamora et al., now suitable for publication in “Biomolecules”.
Author Response
Thank you very much for your positive comments on our article.
Reviewer 4 Report
Comments and Suggestions for Authors
No comments
Author Response
Thank you for the useful revision.